# LIPSCHITZ BANDITS IN OPTIMAL SPACE

**Xiaoyi Zhu**
School of Data Science
Fudan University
Shanghai, China
zhuxy22@m.fudan.edu.cn

**Zengfeng Huang** *
School of Data Science
Fudan University
Shanghai, China
huangzf@fudan.edu.cn

## ABSTRACT

This paper considers the Lipschitz bandit problem, where the set of arms is continuous and the expected reward is a Lipschitz function over the arm space. This problem has been extensively studied. Prior algorithms need to store the reward information of all visited arms, leading to significant memory consumption. We address this issue by introducing an algorithm named Log-space Lipschitz bandits (Log-Li), which achieves an optimal (up to logarithmic factors) regret of $\widetilde{O}\left(T^{\frac{d_z+1}{d_z+2}}\right)$ while only uses $O\left(\log T\right)$ bits of memory. Additionally, we provide a complexity analysis for this problem, demonstrating that $\Omega\left(\log T\right)$ bits of space are necessary for any algorithm to achieve the optimal regret. We also conduct numerical simulations, and the results show that our new algorithm achieves regret comparable to the state-of-the-art while reducing memory usage by orders of magnitude.

## 1 INTRODUCTION

In a multi-armed bandit (MAB) problem, an online algorithm must select from a set of strategies over a sequence of $n$ trials, with the objective of maximizing the cumulative payoff of the chosen strategies. These problems form the fundamental theoretical framework for analyzing the trade-offs between exploration and exploitation that are intrinsic to sequential decision-making under conditions of uncertainty. Algorithms and methodologies for these problems find successful applications in areas such as online auctions, adaptive routing, and the theory of learning in games (Silver et al., 2016; Schneider & Zimmert, 2024).

In this paper, we study the Lipschitz bandit problem - a specific type of MAB problem where the expected reward is Lipschitz continuous(Kleinberg et al., 2008). Existing Lipschitz bandit algorithms heavily rely on storing the reward statistics (e.g. the number of pulls and the mean reward of an arm observed so far) for the visited arms in the memory. Due to the intrinsic assumption in Lipschitz bandit problems that arms belong to a large set, these algorithms need to store information for $poly(T)$ arms, leading to significant space consumption. For instance, the Zooming algorithm (Kleinberg et al., 2008) and Hierarchical Optimistic Optimization (HOO) algorithm (Bubeck et al., 2011) require $O(T)$ space, while the state-of-the-art A-BLiN (Feng et al., 2022) has a space complexity of $O\left(T^{\frac{d_z+1}{d_z+2}}\left(\log T\right)^{-\frac{d_z+1}{d_z+2}}\right)$($d_z$ is the zooming dimension). These space requirements are impractical in many real-world scenarios with high time horizons $T$, leading to a natural question:

*What is the memory cost of achieving optimal regret for the Lipschitz bandit problem?*

Recently, learning with constrained memory has garnered significant attention due to its myriad practical applications. Extensive research has been conducted in various fields, including kernel methods(Williams & Seeger, 2000), convex optimization(Marsden et al., 2022; Blanchard et al., 2023), and general machine learning algorithms(Mitliagkas et al., 2013). Memory constraints have also been explored in the context of multi-armed bandit (MAB) settings, though most studies have focused on stochastic MAB in a streaming setting (Maiti et al., 2021; Jin et al., 2021; Assadi & Wang, 2024). These approaches, however, do not apply to the Lipschitz bandit problem, which involves an infinite set of arms without a stream.

---

*Corresponding Author; also affiliated with SII.

To solve the problem, we present a novel algorithm, named Log-space Lipschitz(Log-Li) that only uses a memory of $O\left(\log T\right)$ bits space while achieving an optimal regret rate of $\widetilde{O}\left(T^{\frac{d_z+1}{d_z+2}}\right)$. Log-Li maintains the empirically best arm and strategically narrows the arm set to learn regions of high reward. This suggests the main finding in this paper that, surprisingly, there is almost no tradeoff between regret and space efficiency for the Lipschitz bandit problem. We further complement this result with a lower-bound complexity analysis and show that $\Omega\left(\log T\right)$ is necessary for achieving optimal regret rate. This means that Log-Li is optimal for both space complexity and regret rate.

## 1.1 Problem Definitions and Preliminaries

We consider a compact doubling metric space $(\mathcal{A}, d_\mathcal{A})$ of arms. Each arm is associated a sub-Gaussian distribution where the expected reward $\mu : \mathcal{A} \to \mathbb{R}$ satisfies the 1-Lipschitz condition that $|\mu(x_1) - \mu(x_2)| \le d_\mathcal{A}(x_1, x_2)$ for any $x_1, x_2 \in \mathcal{A}$. For simplicity, we assume the parameter for the subgaussian distribution to be 1, as extending to other value is straightforward.

Let $\mu^\star = \max_{x \in \mathcal{A}} \mu(x)$ denote the maximal expected payoff. The objective of the algorithm is to minimize regret, defined as $R(T) = \sum_{t=1}^{T} (\mu^\star - \mu(x_t))$. For convenience, we define the optimal gap of $x$ as $\Delta_x = \mu^\star - \mu(x)$ for all $x \in \mathcal{A}$.

Consistent with other works, we restrict our consideration to the metric space $\left([0, 1]^d, \|\cdot\|_\infty\right)$ for simplicity, since we can always embed a doubling space into a Euclidean space with some distortion of metric by the Assouad's embedding theorem (Assouad, 1983). The ideas underlying our algorithms can be generalized to other doubling metric spaces.

In accordance with previous work, we define the zooming dimension as follows.

**Definition 1.** *For a problem instance with arm set $\mathcal{A}$, metric $d_\mathcal{A}$ and expected payoff $\mu$. Let $X_r = \{x \in \mathcal{A} : \Delta_x = \mu^* - \mu(x) \le r\}$. We define the $r$-zooming number as $N_r = \mathcal{N}\left(X_{16r}, \frac{r}{2}\right)$, which is the $\frac{r}{2}$-packing number of $X_{16r}$. The zooming dimension is then defined as*

$$d_z := \min\left\{d \ge 0 : \exists a > 0, N_r \le ar^{-d}, \forall 0 < r < 1\right\}$$

*Moreover, we define the zooming constant $C_z$ as*

$$C_z = \min\left\{a > 0 : N_r \le ar^{-d_z}, \forall 0 < r < 1\right\}$$

The zooming dimension $d_z$ can be significantly smaller if the set of near-optimal arms is "small" in terms of the packing number.

## 1.2 Main Results

This paper investigates the space complexity for the Lipschitz bandit problem and offers both upper and lower bounds.

**Space upper bounds.** Our main contribution is the introduction of the Log-Li algorithm, which solves Lipschitz bandits using logarithmic space. With the Doubling Edge-length Sequence $r_m = 2^{-m+1}$, Log-Li achieves the optimal regret rate of $\widetilde{O}\left(T^{\frac{d_z+1}{d_z+2}}\right)$ using $O(\log T)$ bits.

**Theorem 1.** *With probability exceeding $1 - 2\delta$, the $T$-step total regret $R(T)$ of Log-Li with Doubling Edge-length Sequence $r_m = 2^{-m+1}$ satisfies*

$$R(T) \le (512C_z + 16) \cdot T^{\frac{d_z+1}{d_z+2}} (\log(T/\delta))^{\frac{1}{d_z+2}},$$

*where $d_z$ is the zooming dimension of the problem instance. Moreover, the space complexity of Log-Li is $O(\log T)$ bits.*

The details of the algorithm can be found in Algorithm 2 and Algorithm 1. The proof of Theorem 1 can be found in Section 3. The core of bandit algorithms lies in balancing exploration and exploitation. Prior research addressed this balance by storing the results of all explored arms. This approach allowed their algorithms to deactivate undesirable regions, thereby limiting the number of pulls in these areas and enabling continued exploration in promising directions.

Log-Li, however, adopts a different strategy to reduce space consumption. Instead of deactivating suboptimal regions entirely, it allows revisits to less favorable areas. The frequency of these revisits is controlled based on each area's contribution to regret. With careful analysis, the regret rate of Log-Li is shown to be $\widetilde{O}\left(T^{\frac{d_z+1}{d_z+2}}\right)$.

In contrast, seminal studies (Kleinberg et al., 2008; Bubeck et al., 2011; Feng et al., 2022) demonstrate that the optimal regret bound for Lipschitz bandits without space limitations is $R(T) \lesssim T^{\frac{d_z+1}{d_z+2}} \cdot (\log T)^{\frac{1}{d_z+2}}$. Consequently, Log-Li attains the optimal regret rate for Lipschitz bandits while utilizing minimal space.

**Space lower bounds.** Our next contribution is the space lower bounds for the Lipschitz bandit.

**Theorem 2.** *Consider Lipschitz bandit problems with time horizon $T$ larger than a large enough constant and zooming dimension $d_z \leq d$. Then for any algorithm, if the regret bound $\mathbb{E}[R(T)] \leq \frac{1}{12} T^{\frac{d_z+1}{d_z+2}}$, we must have that the space complexity of the algorithm is at least $\frac{1}{2}\log T$ bits.*

Consequently, Log-Li algorithm is optimal in terms of both regret and space. We also conduct numerical simulations in Section 5, and the results show that our Log-Li achieves regret comparable to the state-of-the-art while reducing memory usage by orders of magnitude.

### 1.3 OTHER RELATED WORK

**Prior work on MAB.** The history of the Multi-Armed Bandit (MAB) problem dates back to Thompson (1933), with a significant surge of activity in recent decades. Several notable algorithms, such as UCB (Agrawal, 1995), the arm elimination strategy(Even-Dar et al., 2006; Perchet & Rigollet, 2013), the $\varepsilon-$greedy method(Auer, 2002), the exponential weights and mirror descent frameworkAuer et al. (2002), have been shown to achieve order-optimal cumulative regret. The Lipschitz bandit problem was first introduced in Kleinberg et al. (2008) and holds significant importance on its own. The Zooming algorithm (Kleinberg et al., 2008) and the Hierarchical Optimistic Optimization (HOO) algorithm (Bubeck et al., 2011) were developed for general doubling metric space.

**Learning under limited space.** Recently, there has been a surge of work on understanding learning under information constraints such as limited memory or communication constraints. One line of research follows the breakthrough paper of Raz (2017), which shows that any learning algorithm for parity problem requires either a memory of quadratic size or an exponential number of samples. Subsequent works have extended these techniques to other learning problems, such as linear regression (Sharan et al., 2019) and noisy version of the parity problem (Garg et al., 2021). There is also significant work on memory lower bounds for random-order streaming models, addressing problems such as entropy estimation (Acharya et al., 2019) and the needle problem (Lovett & Zhang, 2023).

**MAB under limited space.** Most works focus on the (stochastic) multi-arm bandit problem in the streaming setting where both regret minimization and pure exploration are studied. The streaming pure exploration MAB was first introduced by Assadi & Wang (2020), and algorithms are proposed to find $\varepsilon$-best arms with $O(1)$ memory and $O\left(\frac{K}{\varepsilon^2}\right)$ pulls. These algorithms were further developed by Assadi & Wang (2022) to achieve instance-optimal sample complexity. For the regret minimization problem, building on earlier algorithms by Liau et al. (2018); Chaudhuri & Kalyanakrishnan (2020), Agarwal et al. (2022) provided upper and lower regret bounds that are tight in $T$. The lower bound was further improved by Li et al. (2023). Additionally, there is another line of work focusing on the closely related expert learning problem (Srinivas et al., 2022; Peng & Zhang, 2023).

## 2 ALGORITHM

With space constraints, the agent only has knowledge about a constant number of arms in the environment. To fully learn the landscape of the reward, the agent must keep exploring different regions of the arm set. Since the agent forgets information about most regions, it may revisit areas that have previously been eliminated, leading to sub-optimal regrets. To overcome this problem, we gradually partition the regions of the arm set and perform exploration in an iterative deepening manner.

Log-Li operates based on finite partitions of the arm space. To keep exploring while avoiding consistent visits to suboptimal areas, we control the maximum depth the algorithm can search in each

round. The algorithm must start from depth 1 each round, as it has already forgotten which areas were previously eliminated. This process may cause redundant visits to undesirable regions. To mitigate this problem, the algorithm maintains the empirically best arm in the last round and uses it to perform arm elimination. This strategy prevents us from searching too deeply in suboptimal areas. By carefully designing partitions and determining the number of pulls for each round, we can effectively control the overall regret. The learning process is summarized in Algorithm 1 and 2.

---

**Algorithm 1:** Logarithmic Space Lipschitz for Each round (RoundFunc)

---

**Input:** Time horizon $T$; current time $t$; maximum depth $m$; current depth $h$; current cube $C$; comparison arm reward $\tilde{\mu}^{m-1}$; current round max reward $\tilde{\mu}^m$.

**if** $t + n_h > T$ **then**
  $\quad \llcorner$ **return**

Play arm $x_{C,1}^m, \cdots, x_{C,n_h}^m$ from $C$ . Collect the rewards of these pulls $y_{C,1}^m, \cdots, y_{C,n_h}^m$ and
  compute the average payoff $\hat{\mu}_h^m(C) = \frac{\sum_{i=1}^{n_h} y_{C,i}^m}{n_h}$.

**if** $h$ *equals to* $m$ **then**
  $\quad \llcorner$ Compute $\tilde{\mu}^m = \max\{\tilde{\mu}^m, \hat{\mu}_h^m(C)\}$.

**else**
  $\quad$ **if** $\tilde{\mu}^{m-1} - \hat{\mu}_h^m(C) < 4r_h$ **then**
  $\quad\quad$ Equally partition current cube into $(r_h/r_{h+1})^d$ subcubes and define $\mathcal{B}$ as the collection
  $\quad\quad$ of these subcubes.
  $\quad\quad$ **foreach** *subcube* $B \in \mathcal{B}$ **do**
  $\quad\quad\quad \llcorner$ RoundFunc$(T, t + n_h, m, h + 1, B, \tilde{\mu}^{m-1}, \tilde{\mu}^m)$.

  $\quad$ **else**
  $\quad\quad \llcorner$ Eliminate the cube $C$ implicitly by ending the recursion.

---

**Algorithm 2:** Logarithmic Space Lipschitz(Log-Li)

---

**Input:** Arm set $\mathcal{A} = [0,1]^d$; time horizon $T$.

**Initialize:** Error probability $\delta$; Number of rounds $B$; Edge-length sequence $\{r_m\}_{m=1}^{B+1}$; current
  time $t = 0$; comparison arm reward $\tilde{\mu}^0 = 0$

Compute $n_m = \frac{16\log(T/\delta)}{r_m^2}$ for $m = 1, 2, \cdots, B$.

**for** $m = 1, 2, \cdots, B$ **do**
  $\quad$ Equally partition $\mathcal{A}$ to $\left(\frac{1}{r_1}\right)^d$ subcubes and define $\mathcal{B}$ as the collection of these subcubes.
  $\quad$ Set $\tilde{\mu}^m = 0$.
  $\quad$ **for** *each subcube* $B \in \mathcal{B}$ **do**
  $\quad\quad \llcorner$ RoundFunc$(T, t, m, 1, B, \tilde{\mu}^{m-1}, \tilde{\mu}^m)$.

**Cleanup:** play the arm corresponding to $\tilde{\mu}^{m-1}$ until all $T$ steps are used.

---

A running example of Log-Li can be found in figure 1. In each round, Log-Li starts from depth 1 and re-examines the rewards of the cubes. By leveraging the empirically best reward from the last round, we can eliminate suboptimal areas (dark cubes) at the same depth as the previous round without significantly impacting our regret. Simultaneously, we collect rewards from the promising regions (white cubes) for further exploration.

It is worth mentioning that the partition process in the algorithm is detailed explicitly for clarity and comprehension. In practice, the algorithm only requires iterating over the newly partitioned subcubes sequentially, without the need to explicitly store information for each individual subcube.

Moreover, The overhead running time of the algorithm is $O(T)$ since at each time stamp we simply collect a reward and compare it with $\tilde{\mu}^m$. In comparison, the time complexity of the Zooming Algorithm (Kleinberg et al., 2008) is $O(T^2)$, while the time complexity of the HOO algorithm(Bubeck et al., 2011) is $O(T \log T)$.

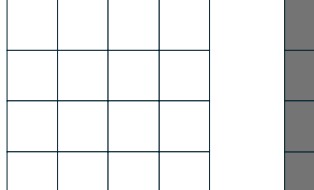 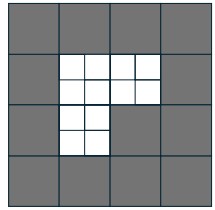 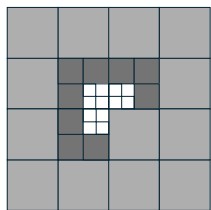

Figure 1: Runing example of Log-Li. The $i$th subfigure reflects the results after the $i$th round. The dark cubes are the eliminated areas and the white cubes are regions with decent rewards at the maximum depth of the current round.

## 3 SPACE AND REGRET ANALYSIS

In this section, we provide space regret analysis for Log-Li. The highlight of the finding is that $O(\log T)$ bits are sufficient to achieve optimal regret rate of $\widetilde{O}\left(T^{\frac{d_z+1}{d_z+2}}\right)$ summarized in Theorem 1.

### 3.1 SPACE ANALYSIS

We would start with the space analysis which is more straightforward.

**Lemma 1.** *Algorithm 1 and 2 consumes $O(\log T)$ bits.*

*Proof.* Note that at each timestamp, Algorithm 1 and 2 only needs to store:

- The id and the empirical mean of the best arm in the last round $\tilde{\mu}^{m-1}$.
- The id and the empirical mean of the best arm in the current round $\tilde{\mu}^m$.
- The id and the empirical mean of the current running arm $\hat{\mu}_h^m(C)$.

Since the number of subcubes we may run is bound by $T$, we can use $\log T$ bits to encode the id of each subcube. For the empirical mean, we need to store the number of pulls and the sum of the rewards, which are both bound by $T$ since reward of each pull is less than 1. Therefore, we can encode the empirical mean in $2 \log T$ bits. The total bits consumed by Algorithm 1 and 2 is bound by $3 \times (\log T + 2 \log T) = O(\log T)$ bits.

$\square$

### 3.2 REGRET ANALYSIS

In this subsection, we would prove the regret upper bound of Log-Li with Doubling Edge-length Sequence $r_m = 2^{-m+1}$. Compared to previous research, Log-Li permits revisits to less favorable directions, which may result in additional regret. However, we will manage the number of these revisits by controlling the number of rounds to ensure that the overall regret remains bounded.

Consistent with other studies, we first show that the estimator $\hat{\mu}$ is concentrated to the true expected reward $\mu$ (Lemma 2). In the following analysis, we let $B_{\text{stop}}$ be the total number of rounds and $\mathcal{A}_h$ be the set of cubes we visit when the depth is $h$ in round $m$.

**Lemma 2.** *Define*

$$\mathcal{E} := \left\{ |\mu(x) - \hat{\mu}_h^m(C)| \le r_h + \sqrt{\frac{16 \log(T/\delta)}{n_h}}, \forall 1 \le h \le m \le B_{stop} - 1, \forall C \in \mathcal{A}_h^m, \forall x \in C \right\}.$$

*It holds that $\mathbb{P}(\mathcal{E}) \ge 1 - 2\delta$.*

*Proof.* Fix any cube $C \in \mathcal{A}_h^m$, we have

$$\hat{\mu}_h^m(C) = \frac{\sum_{i=1}^{n_h} y_{C,i}^m}{n_h}, \mathbb{E}\left[\hat{\mu}_h^m(C)\right] = \frac{\sum_{i=1}^{n_h} \mu\left(x_{C,i}^m\right)}{n_h}.$$

Since $\hat{\mu}_h^m(C) - \mathbb{E}\left[\hat{\mu}_h^m(C)\right]$ obeys subgaussian distribution with parameter $\frac{1}{n_h}$. Applying Hoeffding inequality

$$\mathbb{P}\left(\left|\hat{\mu}_h^m(C) - \mathbb{E}\left[\hat{\mu}_h^m(C)\right]\right| \geq \sqrt{\frac{16 \log(T/\delta)}{n_h}}\right) \leq \frac{2\delta}{T^8}.$$

By the Lipschitz condition of $\mu$ that $\left|\mathbb{E}\left[\hat{\mu}_h^m(C)\right] - \mu(x)\right| \leq r_h, \forall x \in C$, we have

$$\mathbb{P}\left(\sup_{x \in C} |\mu(x) - \hat{\mu}_h^m(C)| \leq r_h + \sqrt{\frac{16 \log(T/\delta)}{n_h}}\right) \geq 1 - \frac{2\delta}{T^8}.$$

Any cube $C \in \mathcal{A}_h^m$ is played for not less than 1 time, and thus $|\mathcal{A}_h^m| \leq T$. On the other hand, there are at most $B_{stop} \leq T$ rounds. Taking a union bound over $C \in \mathcal{A}_h^m$ and $1 \leq h \leq m \leq B_{\text{stop}}$, we would have that $\mathcal{E}$ holds with probability at least $1 - 2\delta$. □

We then show that the optimal arm survives all eliminations with high probability (Lemma 3).

**Lemma 3.** *Under event $\mathcal{E}$ (defined in Lemma 2), the optimal arm $x^* = \arg\max \mu(x)$ is not eliminated in any round.*

*Proof.* Fix any round $m \in [B]$. We use $(C_h^m)^*$ to denote the cube containing $x^*$ in $\mathcal{A}_h^m$. We will show that $(C_h^m)^*$ is not eliminated in depth $h$ for $1 \leq h \leq m - 1$. Under event $\mathcal{E}$, for any cube $C \in \mathcal{A}_{m-1}^{m-1}$ and $x \in C$, we have

$$\hat{\mu}_{m-1}^{m-1}(C) - \hat{\mu}_h^m\left((C_h^m)^*\right) \leq \mu(x) + \sqrt{\frac{16 \log(T/\delta)}{n_{m-1}}} + r_{m-1} - \mu(x^*) + \sqrt{\frac{16 \log(T/\delta)}{n_h}} + r_h$$

$$\leq 2r_{m-1} + 2r_h \leq 4r_h,$$

where the second inequality holds for $n_m = \frac{16 \log(T/\delta)}{r_m^2}$ and the last inequality holds for decreasing property of Doubling Edge-length Sequence. By the elimination rule, $(C_h^m)^*$ is not eliminated. □

Based on lemma 2 and 3, we can control the loss of cubes in each depth.

**Lemma 4.** *Under event $\mathcal{E}$ (defined in Lemma 2), for any $1 \leq h \leq m \leq B_{stop}$, any $C \in \mathcal{A}_h^m$ and any $x \in C, \Delta_x$ satisfies*

$$\Delta_x \leq 8r_{h-1}$$

*Proof.* We again fix any round $m \in [B]$. For $h = 1$, this is straightforward from the Lipschitz condition of $\mu$. For $h > 1$, let $(C_{m-1}^{m-1})^*$ be the cube in $\mathcal{A}_{m-1}^{m-1}$ such that $x^* \in (C_{m-1}^{m-1})^*$. This cube $(C_{m-1}^{m-1})^*$ is well-defined under $\mathcal{E}$ by Lemma 3. For any cube $C \in \mathcal{A}_h^m$ and $x \in C$, it is obvious that $x$ is also in the parent of $C$ (the cube in the previous depth that contains $C$), denoted as $C_{\text{par}}$. For any $x \in C$, we have

$$\Delta_x = \mu^* - \mu(x) \leq \hat{\mu}_{m-1}^{m-1}\left((C_{m-1}^{m-1})^*\right) + \sqrt{\frac{16 \log(T/\delta)}{n_{m-1}}} + r_{m-1} - \hat{\mu}_{h-1}^m(C_{\text{par}}) + \sqrt{\frac{16 \log(T/\delta)}{n_{h-1}}} + r_{h-1}.$$

Due to our choice of $n_m = \frac{16 \log(T/\delta)}{r_m^2}$, we have

$$\Delta_x \leq \hat{\mu}_{m-1}^{m-1}\left((C_{m-1}^{m-1})^*\right) - \hat{\mu}_{h-1}^m(C_{\text{par}}) + 2r_{m-1} + 2r_{h-1}$$

$$\leq \hat{\mu}_{m-1}^{m-1}\left((C_{m-1}^{m-1})^*\right) - \hat{\mu}_{h-1}^m(C_{\text{par}}) + 4r_{h-1},$$

where the last inequality holds from the decreasing property of Doubling Edge-length Sequence. It is obvious that $\hat{\mu}_{m-1}^{m-1}\left(\left(C_{m-1}^{m-1}\right)^*\right) \leq \tilde{\mu}^{m-1}$. Since the cube $C_{\text{par}}$ is not eliminated, we can easily derive the following from the elimination rule

$$\Delta_x \leq 4r_{h-1} + 4r_{h-1} \leq 8r_{h-1}.$$

□

We are now ready to prove Theorem 1.

*Proof of Theorem 1.* Let $R_m$ denote regret of the $m$-th round. Fixing any positive number $B$, the total regret $R(T)$ can be divided into two parts: $R(T) = \sum_{m \leq B} R_m + \sum_{m > B} R_m$. In the following, we bound two parts separately and then determine $B$ to obtain the upper bound of the total regret.

We firstly fix any round $m$. Recall that $\mathcal{A}_h^m$ is set of the survived cubes in depth $h$. According to Lemma 4 , for any $x \in \cup_{C \in \mathcal{A}_h^m} C$, we have $\Delta_x \leq 8r_{h-1} = 16r_h$. Each cube in $\mathcal{A}_h^m$ is a $\|\cdot\|_\infty$-ball with radius $\frac{r_h}{2}$, and is a subset of $S(16r_h)$. Therefore, $\mathcal{A}_h^m$ forms a $\left(\frac{r_h}{2}\right)$-packing of $S(16r_h)$, and the definition of zooming dimension yields that

$$|\mathcal{A}_h^m| \leq N_{r_h} \leq C_z r_h^{-d_z} \leq C_z 2^{(h-1)d_z},$$

where the last inequality holds since $r_h = 2^{-h+1}$. The total regret of the $m$-th round is

$$R_m = \sum_{h=1}^{m} \sum_{C \in \mathcal{A}_h^m} \sum_{i=1}^{n_h} \Delta_{x_{C,i}^m} \leq \sum_{h=1}^{m} |\mathcal{A}_h^m| \cdot \frac{16 \log(T/\delta)}{r_h^2} \cdot 16r_h$$

$$\leq \sum_{h=1}^{m} C_z 2^{(h-1)d_z} \cdot \frac{256 \log(T/\delta)}{r_h} = \sum_{h=1}^{m} C_z 2^{(h-1)(d_z+1)} \cdot 256 \log(T/\delta).$$

Therefore, we have

$$\sum_{m \leq B} R_m = \sum_{m \leq B} \sum_{h=1}^{m} C_z 2^{(h-1)(d_z+1)} \cdot 256 \log(T/\delta)$$

$$= \sum_{m \leq B} (B - m + 1) C_z 2^{(m-1)(d_z+1)} \cdot 256 \log(T/\delta).$$

By standard calculation, we should have that for any real number $a$,

$$\sum_{m \leq B} (B - m + 1) a^m$$

$$= \sum_{m \leq B} (B + 1) a^m - \sum_{m \leq B} m a^m = (B + 1) \sum_{m \leq B} a^m - \left(\frac{B}{a-1} a^{B+1} - \frac{1}{a-1} \sum_{m \leq B} a^m\right)$$

$$= \left(B + 1 + \frac{1}{a-1}\right) \cdot \sum_{m \leq B} a^m - \frac{B}{a-1} a^{B+1} \leq \left(B + 1 + \frac{1}{a-1}\right) \cdot \frac{a^{B+1}}{a-1} - \frac{B}{a-1} a^{B+1}$$

$$= \left(1 + \frac{1}{a-1}\right) \cdot \frac{a^{B+1}}{a-1} \leq 2a^B,$$

which gives that

$$\sum_{m \leq B} R_m = 256 C_z \log(T/\delta) \cdot 2^{-(d_z+1)} \sum_{m \leq B} (B - m + 1) \left(2^{(d_z+1)}\right)^m$$

$$\leq 512 C_z \log(T/\delta) \cdot 2^{(B-1)(d_z+1)}.$$

On the other hand, Lemma 4 implies that the arm corresponding to $\tilde{\mu}^{B-1}$ must satisfy that $\Delta_x \leq 8r_{B-1} = 16r_B = 16 \cdot 2^{-B+1}$. Therefore, we finally have

$$R(T) = \sum_{m \leq B} R_m + \sum_{m > B} R_m \leq 512 C_z \log(T/\delta) \cdot 2^{(B-1)(d_z+1)} + 16 \cdot 2^{-B+1} T.$$

This inequality holds for any positive $B$. By choosing $B^* = 1 + \frac{\log \frac{T}{\log(T/\delta)}}{d_z + 2}$, we have

$$R(T) \le (512C_z + 16) \cdot T^{\frac{d_z+1}{d_z+2}} (\log(T/\delta))^{\frac{1}{d_z+2}}.$$

$\square$

## 4 SPACE LOWER BOUND

In this section, we present space lower bounds for Lipschitz bandit problem as summarized in Theorem 2. Essentially, Theorem 2 says that if we want to achieve the optimal regret bound, we must consume at least $\Omega(\log T)$ bits. Similar to most works, we would first construct problem instances that are difficult to differentiate and then prove lower bounds on these instances.

### 4.1 HARD CASE

We would construct a set of problem instances that are difficult to distinguish. Let $r = \frac{1}{T^{\frac{1}{d_z+2}}}$ and $K = \frac{c_z}{r^{d_z}} = c_z T^{\frac{d_z}{d_z+2}}$. Here $c_z$ is a small enough constant satisfying by the definition of zooming dimension, we can find a set of arms $\mathcal{U} = \{u_1, \cdots, u_K\}$ such that $d(u_i, u_j) \ge r$ for any $i \ne j$. Then we consider a set of problem instances $\mathcal{I} = \{I_1, \cdots, I_K\}$. The expected reward for $I_i$ is

$$\mu_i(x) = \begin{cases} \frac{1}{2} + r, & \text{if } x = u_i, \\ \max\left\{\frac{1}{2}, \mu_i(u_i) - d(x, u_i)\right\}, & \text{otherwise.} \end{cases}$$

Note that here we would have that $\log K = \frac{d_z}{d_z+2} \log c_z \log T = \Theta(\log T)$.

### 4.2 PROOF OF THEOREM 2

We consider the "best-cube identification" problem that after $t$ rounds, the algorithm outputs a cube $y_t$: a prediction for which cube is optimal (has the highest mean reward). For this problem, we may find that if our space is small, the prediction quality will be very poor.

**Lemma 5.** *Consider "best-cube identification" problem with $K \ge 10$ and space complexity less than $\frac{1}{2} \log(K)$. Then there exists at least $\lceil \frac{K}{3} \rceil$ cubes that, for problem instances in Section 4.1,*

$$\Pr\left[y_t = i \mid \mathcal{I}_i\right] < \frac{3}{4}.$$

*Proof.* We consider that summing up all the instance cases and use $M$ to denote the memory state.

$$\sum_{i=1}^{K} \Pr\left[y_t = i \mid \mathcal{I}_i\right] = \sum_{i=1}^{K} \sum_{m} \Pr\left[y_t = i \mid M = m, \mathcal{I}_i\right] \Pr\left[M = m \mid \mathcal{I}_i\right]$$

$$= \sum_{i=1}^{K} \sum_{m} \Pr\left[y_t = i \mid M = m\right] \Pr\left[M = m \mid \mathcal{I}_i\right] \le \sum_{i=1}^{K} \sum_{m} \Pr\left[y_t = i \mid M = m\right]$$

$$= \sum_{m} \sum_{i=1}^{K} \Pr\left[y_t = i \mid M = m\right] = \sum_{m} 1 \le 2^{\frac{1}{2} \log(K)} = K^{\frac{1}{2}},$$

where the second equation holds since the prediction is determined by the memory state, and the last inequation is due to our bound of space complexity. To prove the lemma, assume for contradiction that we have more than $\frac{2K}{3}$ cubes with $\Pr\left[y_t = i \mid \mathcal{I}_i\right] \ge \frac{3}{4}$. We would have that

$$\sum_{i=1}^{K} \Pr\left[y_t = i \mid \mathcal{I}_i\right] \ge \frac{2K}{3} \times \frac{3}{4} = \frac{1}{2}K > K^{\frac{1}{2}},$$

where the last inequation holds for $K \ge 10$ and leads to the contradiction. $\square$

Lemma 5 immediately implies this corollary.

**Corollary 1.** *Consider a "best-cube identification" problem with $K \geq 10$ and space complexity less than $\frac{1}{2}\log(K)$. Choose a cube $i$ uniformly at random, and run the algorithm on instance $\mathcal{I}_i$. Then $\Pr[y_t \neq i] \geq \frac{1}{12}$, where the probability is over the choice of cube $i$, the randomness in rewards and the algorithm.*

*Proof.* By Lemma 5, we can find a set $\mathcal{A}$ such that

$$\Pr[y_t = i \mid \mathcal{I}_i] < \frac{3}{4}, \forall i \in \mathcal{A}, |\mathcal{A}| \geq \lceil \frac{K}{3} \rceil.$$

Since we choose the cube uniformly at random, we would have that

$$\Pr[y_t \neq i] \geq \frac{1}{K}\sum_{i=1}^{K}\Pr[y_t \neq i \mid \mathcal{I}_i] \geq \frac{1}{K}\sum_{i \in \mathcal{A}}\Pr[y_t \neq i \mid \mathcal{I}_i] \geq \frac{1}{K} \cdot \lceil \frac{K}{3} \rceil \cdot \frac{1}{4} \geq \frac{1}{12}$$

$\square$

Finally, we use Corollary 1 to finish our proof of space complexity.

*Proof of Theorem 2.* For $T$ larger than a large enough constant, we would have that $K = c_z T^{\frac{d_z}{d_z+2}} \geq 10$. Assume for contradiction that we have space complexity less than $\frac{1}{2}\log(K)$. Fix round $t$, Let us interpret the algorithm as a "best-cube identification" algorithm, where the prediction is the closest $u_i$ to the arm $x$ that the algorithm chooses. Note that here choosing the closet $u_i$ will only decrease the regret by our hard instance setting. We can apply Corollary 1 to have that $\Pr[y_t \neq i] \geq \frac{1}{12}$. In words, the algorithm chooses a non-optimal cube with probability at least $\frac{1}{12}$, and choosing a non-optimal cube will incur $\Delta(y_t) = \mu^* - \mu_i(u_{y_t}) = r$ regret. Therefore,

$$\mathbb{E}[R(T)] \geq \sum_{t=1}^{T}\mathbb{E}[\Delta(y_t)] = \sum_{t=1}^{T}\Pr[y_t \neq i] \cdot r \geq \frac{T}{12} \cdot \frac{1}{T^{\frac{1}{d_z+2}}} \geq \frac{1}{12}T^{\frac{d_z+1}{d_z+2}},$$

which leads to contradiction. $\square$

## 5 EXPERIMENTS

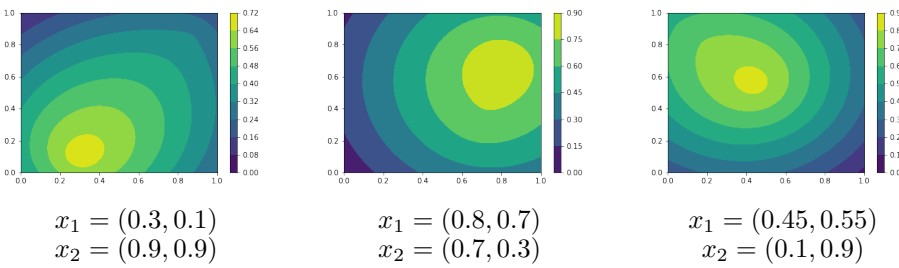

$$x_1 = (0.3, 0.1) \qquad\qquad x_1 = (0.8, 0.7) \qquad\qquad x_1 = (0.45, 0.55)$$
$$x_2 = (0.9, 0.9) \qquad\qquad x_2 = (0.7, 0.3) \qquad\qquad x_2 = (0.1, 0.9)$$

Figure 2: the landscape of $\mu$.

In this section, we evaluate the Log-Li algorithm. In the experiment, the time horizon $T = 100,000$, the arm space is $[0,1]^2$ and the expect reward function is $\mu(x) = 1 - \|x - x_1\|_2 - 0.5\|x - x_2\|_2$ for different values of $x_1$ and $x_2$. The landscape of $\mu$ can be found in figure 2, where the optimal arm is always at $x^\star = x_1$. We select A-BLiN(Feng et al., 2022) and Zooming(Kleinberg et al., 2008) for comparison. Zooming algorithm achieves an optimal regret rate but with a relatively high time complexity of $O(T^2)$ and a space complexity of $O(T)$; A-BLiN delivers performance comparable to the Zooming algorithm while utilizing space proportional to $O\left(T^{\frac{d_z+1}{d_z+2}}(\log T)^{-\frac{d_z+1}{d_z+2}}\right)$ and operating in $O(T)$ time. We will compare the regret rate and space complexity of three algorithms to illustrate the sublinear trend of regret and space efficiency of our algorithm.

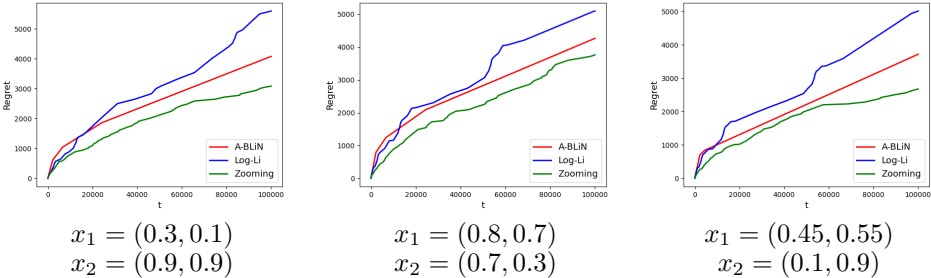

$$x_1 = (0.3, 0.1) \qquad x_1 = (0.8, 0.7) \qquad x_1 = (0.45, 0.55)$$
$$x_2 = (0.9, 0.9) \qquad x_2 = (0.7, 0.3) \qquad x_2 = (0.1, 0.9)$$

Figure 3: The cumulative regret v.s. time horizon $t$.

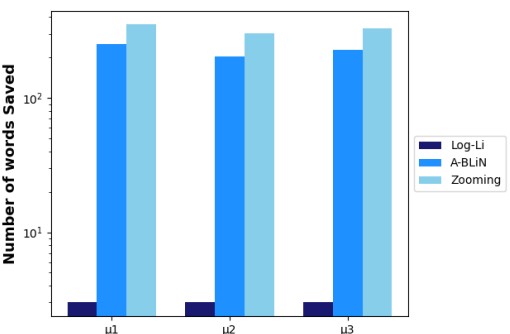

Figure 4: Number of words Saved by Log-Li, A-BLiN and Zooming for Different $\mu$.

Figure 3 illustrates the performance of the three algorithms under different expect reward $\mu$. When $t$ is small, all of the three algorithms are in their initial exploration phases, leading to similar performance. As $t$ increases, the performance of Log-Li tends to be relatively worse compared to the other two algorithms because it restarts the search from depth 1 each time to save space consumption. This could also be a reason why Log-Li experiences increases in regret during the latter stages and exhibits more instability during the exploration process. However, we should also notice that as the search progresses further, Log-Li ultimately achieves a sublinear regret curve, with a growth trend similar to that of A-BLiN. This indicates that the regret rate of Log-Li is asymptotically optimal even with limited space, which is consistent with our analysis in section 3.

We also report the space usage of the three algorithms in figure 4. It is clear that Log-Li significantly reduces the number of arms needed to save and thus consumes much less space. It is noteworthy that this gap will further widen as the time horizon $T$ increases because Log-Li needs to save information about only 3 arms regardless of $T$, whereas the other two algorithms require saving statistics of an increasing number of arms as $T$ grows.

## 6 CONCLUSIONS

In this study, we explore the space complexity for the Lipschitz bandit problem, presenting both upper and lower bounds. Our algorithm, Log-Li, is proven to achieve the optimal regret rate using only $O(\log T)$ bits, whereas prior algorithms require $poly(T)$ bits. This significant improvement in space efficiency greatly reduces memory costs and suggests that there is almost no tradeoff between regret and space efficiency for the Lipschitz bandit problem. Furthermore, the space lower bounds for the Lipschitz bandit indicate that achieving the optimal regret bound necessitates consuming at least $\Omega(\log T)$ bits. Hence, Log-Li is optimal in terms of both regret and space. We also conduct experiments to visualize the performance of our algorithm Log-Li. Our work provides novel insights into designing memory-limited bandit algorithms. It would be valuable to apply the principles of Log-Li to other related bandit problems.

## 7 ACKNOWLEDGEMENTS

This work is supported by National Natural Science Foundation of China No. 62276066, No. U2241212.

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
