# OpenReview forum: "Lipschitz Bandits in Optimal Space"
_ICLR.cc/2025/Conference — ICLR 2025 Poster_

### Official Review · Reviewer_6qq4 · 2024-10-28

**Soundness:** 2
**Presentation:** 3
**Contribution:** 3
**Rating:** 6
**Confidence:** 4

**Summary:**

The paper considers the problem of continuous armed bandit for a reward function which is Lipschitz over the action space. This setting is already well studied in the literature, and optimal regret guarantees are already well-known. The novelty of this work stays in proposing al algorithm which is also efficient on the point of view of the space complexity.
In fact, the authors show both that their algorithm enjoys order $\log(T)$ space complexity (in number of bits) and that this is the lower bound for the setting.

**Strengths:**

The topic of the paper is very interesting. The setting is very interesting, and the space complexity is indeed a limitation of previous approaches for this setting.

The contribution is valuable, and the idea of the paper could bring an impact also outside of the bandit literature.

**Weaknesses:**

Despite I have really appriciated this work, which I think brings a valuable contribution to the bandit community, I am also convinced that this is a little bit immature for a publication at this conference.

I have reached this conlusion after reading many passages of the paper which, in my opinion, need to be deepened.
For example,
 1. In Theorem 1, the result seems to hold with a probability of $1-T^{-5}$. I was rather surprised about this, as most of the other regret bounds in this field fix a given $\delta>0$ and show that the regret is bounded by some function of $T$ multiplied by $\sqrt{\log(1/\delta)}$. At first, I tought this was because the authors were using some given technique which goes outside from usual concentration bounds used in the bandit literature. But in fact, looking at the proof of the result, it seems to me that the proof could be adapted to hold with high probability, as usual. Even if the result is correct, putting the statement in this form could affect the possibility of using this result for future work, and create confusion.
 2. The paper assumes that all arms distribution are Gaussian, even if this assumption is never used except at line 263, where in fact only _subgaussianity_ is used. Again, introducing this (very restrictive) assumption without a reason creates unnecessary confusion in anybody needing to use the results of this paper.
 3. Line 277 "By similar argument to Lemma F.1 in Sinclair et al. (2020) and Lemma 1 in Lu et al. (2019), taking a union bound over C ∈ Amh
and 1 ≤ h ≤ m ≤ Bstop finishes the proof" is not sufficiently precise to be used in a proof
 4. The proof of the space complexity bound seems to ignore the fact that the algorithm has also (according to line 192) to allocate the subcubes and to remember which of theme have already been visitied. By the way the pseudocode is written this seems to take order $2^d$ bits, and it is not clear if it can be done in less space.

In the end, I am not against the acceptance of this work, but I'm wondering if it is in the interest of the author to publish it like this or if it is better to clarify and improve the mentioned passages.

**Questions:**

1. Indeed, to achieve a regret bound of order $T^\alpha$ for any bandit setting, one has to be able to allocate numbers with a precision of at least $T^{\alpha-1}$, as a larger misspecification would compromise the regret bound. This already takes $O(\log(T))$ bits. Does it make the lower bound for the space complexity trivial?

---

> ### Author Response · Authors · 2024-11-21
>
> Thank you for communicating these questions to us. We will clarify them in the following and improve our presentation in the camera-ready version.
>
> __Response to W1__
>
> Thank you for pointing this out. We will adapt the results and proof to hold with high probability in our future versions.
>
> __Response to W2__
>
> Thank you for pointing this out. We do not require the Gaussian assumption; sub-Gaussian conditions are sufficient to meet our requirements. The confusion was caused by a writing error, which we will correct in future versions.
>
> __Response to W3__
>
> Thank you for pointing this out. We will add more explanations of this point in future versions to make the proof clearer.
>
> __Response to W4__
>
> The way we wrote our pseudocode is intended to help readers clearly understand how we partition our set. In practice, our algorithm doesn't need to pre-allocate the subcubes but only needs to traverse them one by one. Since the maximum number of subcubes we will access is T, we only need to maintain the IDs for these T subcubes.
>
> __Response to Q1__
>
> To achieve the desired regret bound, it is indeed necessary to estimate the expected reward with a specified additive error. However, this requirement does not directly imply a space lower bound. Consider the coin problem:  for difference as small as $n^{-0.306}$, there are functions that can amplify this value to a constant using only $O(\log \log n)$ bits [1]. Similarly, in our problem, while precision is important, proving the non-existence of such functions is essential. Our proof take a different aspect by connecting the lower bound with the number of hard instances, which provides a more intuitive understanding and complements our limited-space algorithm.
>
> [1] Braverman M, Garg S, Zamir O. Tight space complexity of the coin problem[C]//2021 IEEE 62nd Annual Symposium on Foundations of Computer Science (FOCS). IEEE, 2022: 1068-1079.

---

> ### Comment · Reviewer_6qq4 · 2024-11-22
> **On the space-complexity lower bound**
>
> I thank the authors for their rebuttal.
>
> The consideration about the coin problem is interesting, but I did not found in the paper the result mentioned by the authors, so I would like to ask them which is it and how does it apply.
>
> Still, about the lower bound if this paper, I try to clarify my reasoning. Let us say that we have just the easier problem of best arm identification, so that it is sufficient to output $x_T$ such that $|x_T-x_*|$ is small.
>
> If we consider the set of reward functions
>
> $$f_a(x)=-|a-x|\qquad x\in (0,1),$$
>
> where the optimal arm of $f_a(\cdot)$ is $a$. We can see that to guarantee $|x_T-x_*|<T^{\alpha-1}$ in every case (this must hold for $\alpha \in (0,1)$, and, being a one-dimensional Lipschitz class precisely with $\alpha=2/3$ to match the lower bound), we need to distinguish at least $T^{1-\alpha}$ instances. This requires at least $\log(T^{1-\alpha})=(1-\alpha)\log(T)$ bits.

---

> > ### Author Response · Authors · 2024-11-23
> >
> > Thank you for your response. The argument regarding the coin problem can be found in section 2.1 of this article, but at the time, I mistakenly thought you meant approximating the reward of keeping an arm.
> >
> > Based on my understanding, your reasoning for the lower bound aligns with the logic of our lower bound proof. In our proof, we first demonstrate that best arm identification requires at least $\Omega(\log k)$ space to distinguish all the hard instances, and then, based on this result, we show that achieving optimal regret also requires at least this much space.
> >
> > In selecting the hard instance, we aimed to provide a tighter bound, resulting in the number of our hard instances being $\Omega\left(T^{\frac{d_z}{d_z+2}}\right)$. This allowed us to ultimately obtain a lower bound of $\Omega(\log K)$ instead of $\Omega((1-\alpha)\log K)$, where the latter would decrease as $d_z$ increases.

---

> > > ### Comment · Reviewer_6qq4 · 2024-11-23
> > > **Thank you, I have no further questions**
> > >
> > > Thank you, I have no further questions

---

### Official Review · Reviewer_pV8P · 2024-10-31

**Soundness:** 2
**Presentation:** 1
**Contribution:** 2
**Rating:** 3
**Confidence:** 3

**Summary:**

This paper proposed a novel algorithm, “Log-space Lipschitz bandits” (Log-Li), which reduces space complexity to $O(\log T)$ from $O(T)$ while achieving near-optimal regret, for the the Lipschitz bandit problem.

**Strengths:**

1. This paper proposed a novel algorithm (Log-Li), which reduces space complexity to $O(\log T)$ from $O(T)$ while achieving near-optimal regret.

2. It shows that the regret achievable with this minimal memory is near optimal.

3. The paper includes theoretical proofs for the space and regret bounds of the algorithm, complemented by experimental results demonstrating that Log-Li maintains regret close to other state-of-the-art algorithms but with significantly lower memory usage.

**Weaknesses:**

There are several weaknesses in the current manuscript.

1. The authors acknowledge that Log-Li’s regret is higher and more variable than the benchmark algorithm A-BLiN (as shown in Fig .3) due to memory limitations, requiring repeated exploration of suboptimal regions.  This means that Log-Li cannot guarantee the regret performance if the required memory decreases.

2.  The evaluation is too weak to support the claim made in this paper. It only compares with one benchmark algorithm and the performance of the proposed Log-Li is even worse than that of the benchmark algorithm.

3. The regret bound is highly dependent on the zooming dimension  $d_z$ . If  d_z  is large, the regret bound  $O(T^{\frac{d_z + 1}{d_z + 2}})$  grows close to linear in $ T$ , suggesting that the Log-Li algorithm may struggle with higher-dimensional spaces. The analysis would benefit from a more in-depth exploration of how the zooming dimension affects both regret and space efficiency in high-dimensional spaces.

4. The presentation of the current manuscript is extremely poor and unfriendly to readers. Though this reviewer may misunderstand the main technique and theoretical results, the poor presentation is an outstanding issue. From this reviewer's point of view, it is way below the standard.

**Questions:**

My detailed questions are as follows:

1. My first question is about the claimed "optimal regret" in this paper. As indicated in Weakness 3, the regret bound is almost linear with a large zoom dimension $d_z$. If this is the optimal upper bound, does it mean that no algorithm can achieve sublinear regret for Lipschitz bandit in high-dimensional spaces?

2. What is Doubling Edge-Length Sequence  $r_m$? And why do we need to set it as  $2^{-m+1}$?

3. Corollary 1 implies that with  $K \geq 10$ , the probability of non-optimal cube selection exceeds  $\frac{1}{12} $. How would this probability scale as  $T$  grows large? Would the required cube size in terms of $K$ or arm space expand as  $T$ increases, potentially impacting the algorithm’s ability to achieve optimal regret in very high-dimensional spaces?

4. In Lemma 2, you define the event  $E$  to bound the difference between the estimated and true expected reward. However, the analysis assumes that each cube in  $A_h^m$  has been played sufficiently to approximate rewards accurately. Could you clarify how this holds in scenarios with very large or complex arm spaces where  |$A_h^m| $  is substantial? Would the regret increase under limited play-per-cube situations, and if so, how could this impact your regret bound?

5. The paper states that Log-Li’s arm elimination rule requires revisiting previously eliminated areas. In practical terms, this leads to redundant exploration of suboptimal regions, increasing regret. Is there an analytical way to quantify the regret incurred from this repeated exploration? Could adding memory to store limited past elimination history potentially reduce these redundancies?

---

> ### Author Response · Authors · 2024-11-21
>
> Thank you for communicating these questions to us. We will clarify them in the following and improve our presentation in the camera-ready version.
>
> __Response to W1 & W2__
>
> The core of our paper is theoretical analysis. We first prove that the regret of our algorithm, Log-Li, is asymptotically optimal. Based on this, the primary purpose of our experiments is to verify the correctness of our theoretical conclusions, so we focus more on the trend of regret for our algorithm. From our experimental results, it is evident that the regret of our algorithm is indeed asymptotically optimal, which aligns with our expectations. Moreover, our algorithm only uses a constant number of words.
>
> According to our theoretical results, there might be constant differences compared to other algorithms; however, the differences are not significant. Empirically, since we are only concerned with the trend, we have not fine-tuned the parameters of our algorithm. It is possible that by adjusting the parameters, we could achieve better experimental results, but this is not our main focus.
>
> __Response to Q1__
>
> The regret lower bound is proved to be $R(T) = \Omega(T^{\frac{d_z+1}{d_z+2}})$[1]. This lower bound indicates that any algorithm for the Lipschitz bandit problem will incur at least this amount of regret. Therefore, according to our theoretical analysis, the regret of Log-Li is already asymptotically optimal.
>
> [1] A. Slivkins, “Contextual bandits with similarity information,” J. Mach. Learn. Res., vol. 15, pp. 2533–2568, Jan. 2014.
>
> __Response to Q2__
>
> We denote $r_m$ as the edge length because it corresponds to the side length of each subcube in our partition. We describe it as doubling since it decays at a rate of 1/2. We chose this particular edge-length sequence to achieve the desired optimal regret rate in our regret analysis. Due to space constraints, our algorithm inevitably re-explores incorrect directions; using this sequence helps us control the additional regret from these erroneous explorations.
>
> __Response to Q3__
>
> Your first point of view is incorrect and this probability will not scale as T grows large. Our proof for the lower bound of space complexity does not depend on the size of  T.  As demonstrated in our proof, regardless of the size of T, as long as K>10 and space complexity less than $\frac{1}{2}\log K$, our Corollary 1 holds.
>
> As T increases, our cube size decreases and the number of K increases. You can see this specifically in the construction of our hard instance (line 394). This is because as T increases, the allowed error decreases, requiring us to more precisely characterize the landscape of reward and thus increase the space requirement(increase by $\log(T)$).
>
> __Response to Q4__
>
> Your point is incorrect; we do not need each cube to be played sufficiently. In Lemma 2, our estimation range for the true reward is related to the number of times we play each arm, denoted as $n_h$. If the number of plays for a particular cube is small, the estimation range is wider; if the number of plays is large, the range is narrower. Regardless of the number of subcubes, the total number of arm plays is T. Therefore, using the union bound, our Lemma 2 always holds. Consequently, the regret bound we obtain under event E is always valid with high probability.
>
> __Response to Q5__
>
> In the theoretical analysis section, we carefully analyzed all the components of regret caused by Log-Li in each round (line 338). Under our carefully chosen edge sequence, the extra regret incurred is also within the $O(T^{\frac{d_z+1}{d_z+2}})$bound.
>
> Adding memory is definitely helpful because it allows us to retain information about previously explored subcubes, reducing exploration in incorrect directions. However, the main focus of our work is on theoretical analysis, and experimental results are not our primary concern.

---

### Official Review · Reviewer_VDg4 · 2024-11-03

**Soundness:** 3
**Presentation:** 2
**Contribution:** 3
**Rating:** 6
**Confidence:** 3

**Summary:**

This paper introduces a new algorithm for the Lipshitz bandit problem which uses only $O(\log T)$ bits of memory while achieving an optimal regret of $O(T^{\frac{d_{z} + 1}{d_{z} + 2}})$. It also showed that $\Omega(\log T)$ bits of space are necessary for any algorithm to achieve optimal regret. Finally, it numerically showed that the proposed algorithm is superior to an existing method in terms of regret while reducing memory usage sifnificantly.

**Strengths:**

The proposed algorithm named Log-Li (Log-space Lipschitz) algorithm achieves an optimal regret rate in the Lipschitz bandit problem while using a memory of $O(\log T)$ bits space. This is a significant improvement compared to other existing methods, which have to store information of $poly (T)$ arms. They also showed that using $O(\log T)$ bits is unavoidable if we want to achieve an optimal regret rate. From above, the Log-Li algorithm is optimal in terms of regret and space, which is the contribution of this paper.

**Weaknesses:**

- The only algorithm that is compared to the Log-Li algorithm is the A-BLiN algorithm. Even though other existing methods use poly(T) bits of memory space, I believe it is still necessary to empirically compare cumulative regret with them.
- In line 176, Log-Li algorithm generates $(\frac{r_{h}}{r_{h + 1}})^{d}$ subcubes, which possibly consume large amount of memory.
- The log-Li algorithm has a larger cumulative regret than an existing method empirically.

[Missing citation]
- Stefan Magureanu, Richard Combes, and Alexandre Proutiere. Lipschitz Bandits: Regret Lower Bound and Optimal Algorithms. COLT2014.


[Minor comments]
- The authors should refer to the definition of $d_{z}$ in the abstract.
- Periods are missing, and upper and lower case letters are mixed inappropriately.
  - P9 Figure 2-> "The landscape of $\mu$."
  - P10: Figure 4 "Logarithm of number of words saved by Log-Li and A-BLiN for different $\mu$."
  - Line 178:  Space between "for" and "each"
  - Lines 99, 208, 380: "theorem"-> "Theorem "

**Questions:**

- Related to the second weakness above, it seems Log-Li uses a large memory space when we partition the arm set (Line 176). How much memory does the algorithm consume in this procedure? Does it exceed memories than that used to store information of each arm?
- Isn't the "for loop" in Line 194 computationally heavy? In my understanding, the size of $\mathcal{B}$ is exponentially large in $d$, and therefore hurts the computational complexity of the Algorithm.

---

> ### Author Response · Authors · 2024-11-21
>
> We would like to thank the reviewer for communicating these questions to us. We will clarify them in the following and correct the citation and grammar issues in the camera-ready version.
>
> __Response to missing citation & minor comments__
>
> Thank you for pointing that out. We will further refine and edit the text.
>
> __Response to W1 & W3__
>
> The primary focus of our paper is on theoretical analysis. We begin by demonstrating that our algorithm, Log-Li, achieves asymptotically optimal regret. Consequently, the main aim of our experiments is to validate the accuracy of our theoretical findings, which is why we concentrate on observing the trend of regret in our algorithm. Our experimental results clearly show that the regret of our algorithm is asymptotically optimal, confirming our expectations. Furthermore, our algorithm operates using only a constant number of words.
>
> While our theoretical results suggest there may be differences when compared to other algorithms, these differences are not large. Since our empirical focus is on the overall trend, we did not fine-tune the parameters of our algorithm. Although parameter adjustments might yield improved experimental outcomes, this is not our primary concern.
>
> __Response to Q1__
>
> We still present our partitioning in the algorithm this way to make it easier for readers to understand how we perform the partition operation on the set. In the actual algorithm, our algorithm only needs to iterate through the newly partitioned subcubes one by one, without needing to explicitly store information for each of them. Our algorithm only needs to store information about 3 arms, therefore it only requires $O(\log T)$ space.
>
> __Response to Q2__
>
> The "for loop" in Line 194 does not incur significant computational overhead. We explicitly include "for loop"  over all subcubes in our algorithm just to help readers understand our partitioning method. Considering what we do at each time step from 1 to T, we simply collect a reward and occasionally update $\tilde{\mu}^{m}$. These actions do not result in substantial computational overhead.

---

> > ### Comment · Reviewer_VDg4 · 2024-12-01
> >
> > Thank you for the reply. I have no further questions.

---

### Official Review · Reviewer_KiQN · 2024-11-04

**Soundness:** 3
**Presentation:** 3
**Contribution:** 3
**Rating:** 6
**Confidence:** 3

**Summary:**

This paper introduces the Log-space Lipschitz bandits (Log-Li) algorithm to address high memory usage in the Lipschitz bandit problem by achieving optimal regret with only O($\log T$) bits of memory. Next the paper provides a space lower bound for the Lipschitz bandit
problem showing that the algorithm is optimal in terms of both regret and space.

**Strengths:**

- The paper is well written - provides adequate motivation and highlights the technical challenges involved.
- To the best of my knowledge, the proofs in the main paper seem fine.
- The space complexity improvement is significant and makes the solution practical.

**Weaknesses:**

No major weakness.

**Questions:**

- This is regarding the novelty of the analysis. The authors say "Similar to prior research, Log-Li identifies and removes the undesirable region of the arm set while exploring and partitioning the remaining area". Could the authors provide references for these prior research. Further could the authors also provide more details on the novelty of the analysis, how it differs from existing analysis and where, if any have they used modifications of existing results.

---

> ### Author Response · Authors · 2024-11-21
>
> Thank you for communicating these questions to us. We will clarify them in the following and improve our presentation in the camera-ready version.
>
> __Response to Questions__
>
> Since arms belong to a very large set in our problem, it is a classic approach to gradually partition the set to learn the landscape of the reward function, such as in [1,2]. Our algorithm also employs this design, resulting in similarities to past algorithms. In the analysis of previous algorithms, to control the regret caused by exploring in the wrong directions, they managed this by saving all explored arms and promptly eliminating and marking them. This strategy helped limit the number of explorations in unpromising directions. However, this strategy fails in our setting because we can only store information for a constant number of arms and will experience some inevitable repeated exploration in incorrect directions. The novelty of our analysis lies in the control of these extra regrets which is not typically seen in analyses of previous algorithms. By improving our search strategy and making the proper choice of the edge sequence of partitions, we show that even with this additional regret, we can still achieve the desired optimal regret bound.
>
> [1] Robert Kleinberg, Aleksandrs Slivkins, and Eli Upfal. Multi-armed bandits in metric spaces. In Proceedings of the fortieth annual ACM symposium on Theory of computing, pp. 681–690, 2008.
>
> [2] Yasong Feng, Tianyu Wang, et al. Lipschitz bandits with batched feedback. Advances in Neural Information Processing Systems, 35:19836–19848, 2022.

---

> > ### Comment · Reviewer_KiQN · 2024-11-27
> >
> > I thank the authors for anwering my question. I do not have any further questions.

---

### Meta-Review · Area_Chair_hFd1 · 2024-12-21

**Metareview:**

The paper proposes Log-Li, a novel algorithm that achieves near-optimal regret while significantly reducing space complexity to logarithmic levels. Theoretical analyses validate the space and regret bounds, while experimental results show that Log-Li maintains competitive regret compared to state-of-the-art algorithms, albeit with considerably lower memory usage.

However, there are notable weaknesses. Log-Li’s regret is higher and more variable than the benchmark A-BLiN, especially under memory constraints, and its performance degrades in high-dimensional spaces due to dependency on the zooming dimension. I also agree (with Reviewer pV8P) that the presentation of the current manuscript is quite poor as well as the rebuttal is not very comprehensible/ does not address all the comments from the reviewers, e.g., it is not clear how the results and analysis differ from Kleinberg et al. '08. The experimental evaluation is limited to the A-BLiN benchmark, and that, too, the proposed algorithms' regret performance as well as computational efficacy is in question. Based on these, I recommend the authors to please submit the manuscript to the next suitable venue with all the necessary modifications.

**Additional Comments On Reviewer Discussion:**

See above.

---

### Decision · Program_Chairs · 2025-01-22

Accept (Poster)